# Measuring the Depth of Subsurface Defects in Additive Manufacturing Components by Laser-Generated Ultrasound

**Zhixiang Xue [1], Wanli Xu [1], Yunchao Peng [2], Mengmeng Wang [2], Vasiliy Pelenovich [1], Bing Yang [1] and Jun Zhang [1,\*](ID)**

[1] School of Power and Mechanical Engineering, Wuhan University, Wuhan 430072, China; zhixiangxue1119@whu.edu.cn (Z.X.); wanlixu@whu.edu.cn (W.X.); pelenovich@mail.ru (V.P.); toyangbing@whu.edu.cn (B.Y.)

[2] Pipe China Network Corporation Eastern Oil Storage and Transportation Co., Ltd., Xuzhou 221000, China; pengyc02@pipechina.com.cn (Y.P.); wangmm02@pipechina.com.cn (M.W.)

\* Correspondence: zhangjun2010@whu.edu.cn; Tel.: +86-193-7105-7915

**Abstract:** A new method to measure the depth of subsurface defects in additive manufacturing components is proposed based on the velocity dispersion analysis of Lamb waves by the wavelet-transform of laser ultrasound. Firstly, the mode-conversion from laser-generated surface waves to Lamb waves caused by subsurface defects at different depths is studied systematically. Secondly, an additive manufactured 316L stainless steel sample with six subsurface defects has been fabricated to validate the efficiency of the proposed method. The measured result of the defect depth is very close to the real designed value, with a fitting coefficient of 0.98. The defect depth range for high accuracy measurement is suggested to be lower than 0.8 mm, which is enough to meet the inspection of layer thickness during additive manufacturing. The result indicates that the proposed method based on laser-generated ultrasound (LGU) velocity dispersion analysis is robust and reliable for defect depth measurement and meaningful to improve the processing quality and processing efficiency of additive/subtractive hybrid manufacturing.

**Keywords:** additive manufacturing; subsurface defects; laser ultrasound; stainless steel

## 1. Introduction

Metal additive manufacturing (AM) has disruptive applications in many industries, including the aerospace, biomedical, and automotive industries [1]. Compared with traditional manufacturing methods, this layer-by-layer manufacturing technology has many advantages in the customization of products with complex geometric structures [2]. However, mainstream AM methods have interlayer defects such as inclusions and lack-of-fusion buried in the subsurface of the printing layer [3]. To remove the random defects, additive/subtractive hybrid manufacturing is proposed with performing additive and subtractive manufacturing (SM) alternatively until the whole part is fabricated [4]. The online detection and location of defects are indispensable for the SM processing. The more accurate the measurement of the defects' position, the faster SM can repair the defective part. Therefore, the online monitoring method is meaningful to significantly improve processing quality and processing efficiency.

The laser-generated ultrasound (LGU) has been widely used in various manufacturing fields due to its advantages of being non-contact, broadband, and high-resolution [5]. LGU is also considered to be a potential method for the online detection of metal additive manufacturing samples [6]. Current research mainly focuses on the detection of surface and subsurface defects by LGU Rayleigh waves [7]. Zeng Y. produced three kinds of artificial defects including crack, flat bottom hole, and through hole defects and carried out an LGU inspection and finite element analysis on these three kinds of artificial defects [8]. In the defect evaluation, Wang C. used the LGU Rayleigh wave to measure the thickness of

the subsurface defects with rectangular sides. The two ends of the defects were detected separately to quantify the width of the sub-surface groove defects [9]. Chen D. used the phase evolution of LGU Rayleigh waves to detect subsurface defects [10]. Although LGU Rayleigh wave has many advantages in detecting surface and subsurface defects, there are still few applications in measuring defect depth. In a previous finite element analysis, it had been found that ultrasonic surface waves are modulated by near-surface defects, resulting in a waveform conversion from surface waves to Lamb waves. There was a finite element simulation study of surface defects with laser phased array Rayleigh waves [11]. It used the phased array principle to enhance the diffraction wave signal of the LGU detection of cracks and defects [12]. Zhou Z. performed finite element analysis on large-scale surface gaps in LGU inspection and studied the interaction between the Rayleigh wave generated by the laser and surface cracks [13]. Therefore, if we can use appropriate signal processing methods to systematically study the Lamb conversion law, it is possible to propose a defect depth method.

The Lamb waves have velocity dispersion characteristics, which means that the propagation speed of the Lamb wave is changed with frequency, sample thickness, and elastic properties. Based on the principle of velocity dispersion, many researchers focused on the estimation of a material's properties from velocity dispersion analysis using computer-aided signal processing [14]. Previous studies had shown that the attenuation, velocity, frequency, and dispersion characteristics of the Lamb wave generated by the laser are closely related to the anisotropy and viscoelastic properties of the material [15]. Fourier transform and wavelet-transform are two methods to analyze the velocity and dispersion characteristics of Lamb waves. In the application of the Fourier transform, Farouk B. studied the influence of symmetry and discontinuity on the Lamb wave modes [16,17]. This is because of the multi-modal characteristics caused by the velocity dispersion characteristics of the Lamb waves, and it can be quantitatively displayed using the Fourier transform [18]. Although Fourier transform can achieve better results, wavelet-transform has better performance in the field of time-frequency analysis [19]. Amir M. combined wavelet-transform, fast Fourier transform, and modal positioning theory with variable frequency wave speed and considered specific frequency ranges through fast Fourier transform and wavelet packet analysis [20]. In particular, the wavelet-transform enables the transient signal to identify required information and irrelevant information, even overlapping each other in frequency [21–23].

This paper presents a systematic study of the mode-conversion from the LGU surface wave to the Lamb waves caused by subsurface defects at different depths. A new method to measure the depth of subsurface defects is proposed based on the Lamb waves velocity dispersion analysis by wavelet-transform. A 316L stainless steel sample with six subsurface defects is fabricated to validate the efficiency of the proposed method.

## 2. Experimental Setup

The experimental setup is shown in Figure 1. A Nd: YAG pulsed laser (WEDGE 1064HB DB, Pavia, Italy) with a wavelength of 1064 nm and a pulse duration of 12 ns is used to generate ultrasonic waves (Table 1). The Laser receiver (QUARTET-1500 Bossanova, Los Angeles, CA, USA), with an operating wavelength of 532 nm and a bandwidth of 102 MHz, is applied to receive the ultrasonic waves. The stainless steel (316L) plates were fabricated by the selective laser melting method (SLM AmPro SP-500, Victoria, Australia) with 30 mm in length, 5 mm in thickness, and 30 mm in width (Figure 2). A series of notch defects with a fixed area of 3.0 mm × 0.5 mm and varying depths of 0.1, 0.2, 0.3, 0.5, 0.7, and 1 mm were fabricated as subsurface defects in the specimen (Figure 2). In this paper, the pulse laser energy density E can be calculated by $E = (4 \times A \times e) / (\pi d^2)$, where A is the laser coefficient of the sample (here, 0.1 is adopted for A), e is the pulse laser energy with a value of 2 mJ, and d is the Spot diameter with a value of 150 μm. The calculated result of E is 11.3 mJ/cm$^2$, which is significantly lower than the stainless-steel ablation threshold of 450 mJ/cm$^2$ (Table 2) [24]. Therefore, the LGU is controlled by a thermoelastic

mechanism. The sample surface rapidly expands and contracts in the laser heating zone, forming internal stress and strain, which propagates in the form of the elastic wave.

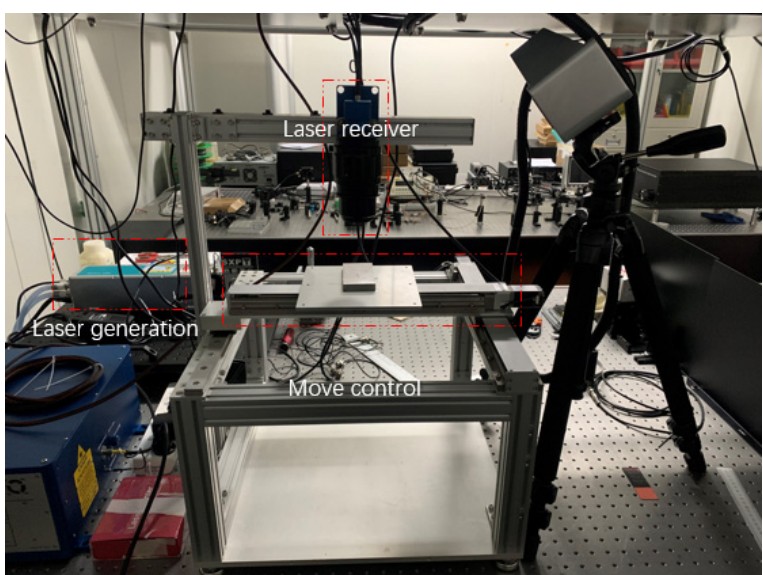

**Figure 1.** LGU testing system.

**Table 1.** WEDGE 1064 HB DB parameters.

| Parameter | Wavelength/(nm) | Laser Coefficient of Sample (A) | Spot Diameter (d)/(µm) | Pulse Energy (e)/(mJ) |
|---|---|---|---|---|
| Value | 1064 | 0.1 | 150 | 2 |

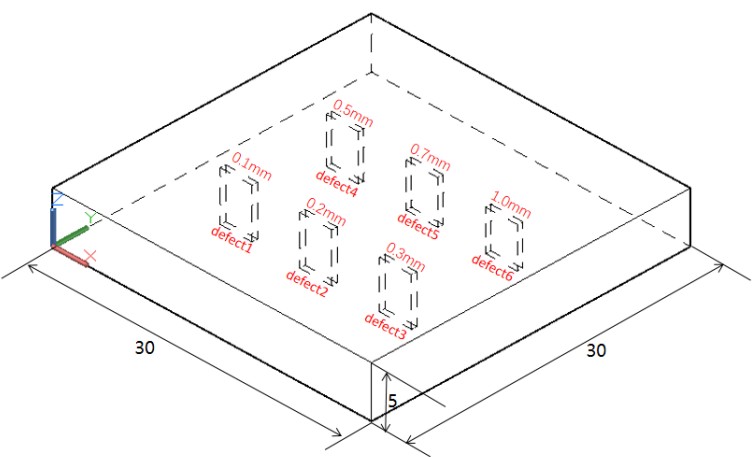

**Figure 2.** Sample schematic with six embedded notches.

**Table 2.** The stainless-steel (316L) parameters.

| Parameter | Longitudinal Sound Velocity $(c_l)$/(m/s) | Shear Wave Velocity $(c_s)$/(m/s) | Rayleigh Wave Velocity $(c_R)$/(m/s) | Ablation Threshold/ (mJ/cm$^2$) |
|---|---|---|---|---|
| Value | 5880 | 3230 | 2990 | 450 |

The schematic of the mechatronic system for generating and detecting the LGU is shown in Figure 3. The laser spots of the excitation and the reception maintains a distance of D, D = 2 mm. The scanning steps $(d_x, d_y)$ are set to 0.1 mm. The acquired A-scans

are arranged and stored into a three-dimensional matrix. The B-scan and C-scan images are plotted by extracting a sub-matrix from the acquisition data, which is helpful to find the horizontal position of the defects. Then, the depth of the defects is measured by the proposed method, explained in the next section.

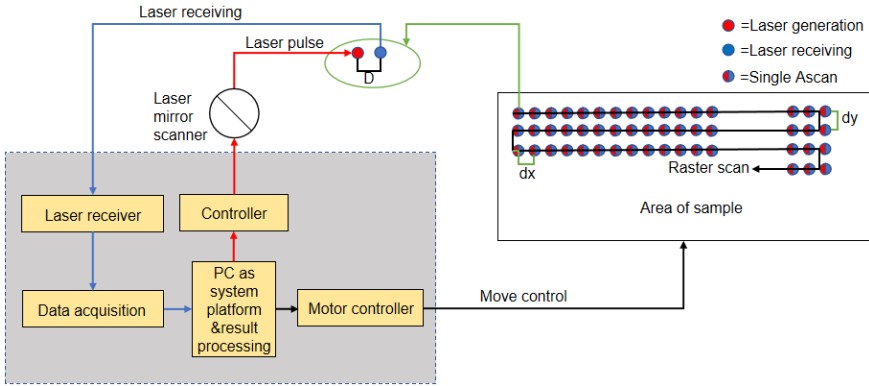

**Figure 3.** LGU system and scanning imaging strategy.

## 3. Method

The proposed method for defect depth measurement, based on the LGU signals, consists of three steps. Firstly, the defect depth is characterized by the phase velocity and central frequency of the Lamb waves based on the velocity dispersion principle. Secondly, time-frequency analysis of the LGU signal is used to obtain its frequency, in which the wavelet-transform is employed. Finally, the velocity of the LGU Lamb waves is calculated with the time of flight from the excitation spots to the reception spots.

### 3.1. Velocity Dispersion of the Lamb Waves

The quantitative relation between the defect depth and the characteristics of the Lamb wave is the key point for the depth measurement. According to the velocity dispersion characteristics of the Lamb wave, the dispersion curves describing the influence of the frequency and velocity on the defect depth could be used for measurement and calibration. The dispersion curves can be calculated by the Rayleigh-Lamb equation of the Lamb wave [25] as follows.

S mode:

$$\frac{tan k_s b}{tan k_l b} = -\frac{4k_0^2 k_l k_s}{\left(k_0^2 - k_s^2\right)^2} \tag{1}$$

A mode:

$$\frac{tan k_s b}{tan k_l b} = -\frac{\left(k_0^2 - k_s^2\right)^2}{4k_0^2 k_l k_s} \tag{2}$$

$$k_l^2 = \left(\frac{\omega}{c_l}\right)^2 - k_0^2 \tag{3}$$

$$k_s^2 = \left(\frac{\omega}{c_s}\right)^2 - k_0^2 \tag{4}$$

$$\omega = 2\pi f \tag{5}$$

Here,

$k_0$—wave number along the horizontal direction of the sample
$b$—1/2 sample thickness
$\omega$—angular frequency
$c_l$—longitudinal wave velocity
$c_s$—shear wave velocity

According to the above equations and the stainless-steel (316L) parameters about $c_l$ and $c_s$ (Table 2), the dispersion curves of A and S mode Lamb waves are shown in Figure 4a. The dispersion curve of the $A_0$ mode is extracted and used for the depth measurement, as shown in Figure 4b. If the velocity and the frequency changes of the Lamb waves induced by the existing defect are measured according to the LGU experiment, the depth of the defect can be calculated by the dispersion curve of the $A_0$ mode.

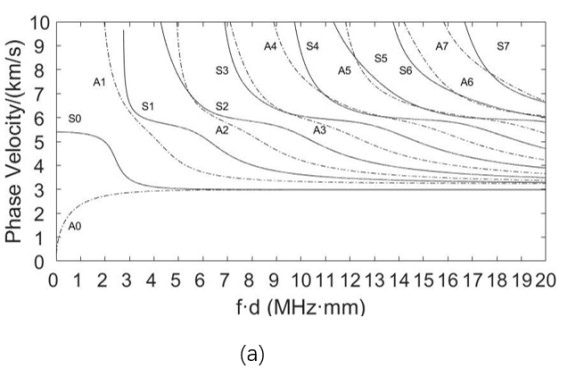
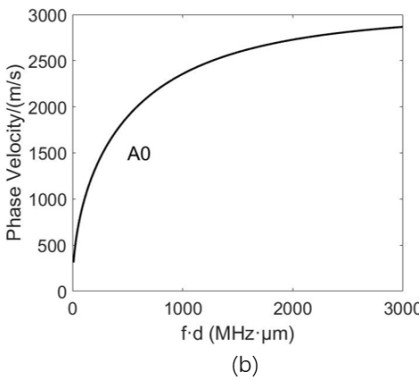

(a)                                                (b)

**Figure 4.** (**a**) Velocity dispersion curves (**b**) $A_0$ mode velocity dispersion curve.

### 3.2. Time and Frequency Measurement by Wavelet-Transform

To get the frequency of the defect signal, a time-frequency analysis based on the wavelet-transform is used. The wavelet-transform of an A-scan signal $f(t)$ is defined as [19]:

$$(\omega_\psi f)(a,b) = \frac{1}{\sqrt{a}} \int_{-\infty}^{+\infty} f(t)\overline{\psi}\left(\frac{t-b}{a}\right) dt \tag{6}$$

Here, $\psi(t)$ designates the basic wavelet, and $\overline{\psi}(t)$ means a complex conjugate. $a$ and $b$ are scale and shift parameters, respectively. $\psi(t)$ must satisfy the admissibility condition by this equation:

$$C_\psi = \int_{-\infty}^{+\infty} \frac{|\psi(\omega)|^2}{|\omega|} d\omega < \infty \tag{7}$$

At the same time, the basic wavelet must satisfy the following two constraints.

$$\int_{-\infty}^{+\infty} |\psi(t)| dt < \infty \tag{8}$$

$$\int_{-\infty}^{+\infty} \psi(t) dt = 0 \tag{9}$$

To obtain a continuous wavelet-transform (CWT), Morlet wavelet is used in this paper; then, the mother wavelet function $\psi(t)$ is expanded and translated by Equations (8) and (9)

$$\psi_{a,b}(t) = \frac{1}{\sqrt{|a|}} \psi\left(\frac{t-b}{a}\right) \quad a,\ b\ \epsilon R.a \neq 0. \tag{10}$$

The generation of $\psi(t)$ depends on the parameters $a$ and $b$. $\psi_{a,b}(t)$ is the wavelet basis function. As the core part of this method, we can perform better time-frequency analysis on the signal through wavelet-transform and extract the time-frequency pairs required by our method to obtain better depth measurement results.

### 3.3. Velocity Calculation

The accurate measurement of time-of-flight is the key point for velocity calculation if the propagation distance is fixed. However, the actual flight time cannot be read directly on the time axis, due to the different system delays in each LGU system. The Rayleigh

wave sound velocity ($C_R$) of 316L stainless steel is used to calculate the actual flight time according to:

$$T_t = d_0/C_R \tag{11}$$

After reading the flight time of the A-scan signal without defects, the system delay can be calculated by subtraction:

$$T_0 = T_t - T \tag{12}$$

$T_t$—Ultrasonic actual flight time
$T$—Ultrasonic flight time read from A-scan
$T_0$—ystem delay r
$d_0$—Fixed distance between generation and receiving end

Then, the phase velocity of the Lamb wave A-scan signal with defects can be calculated:

$$C_p = d_0/(T - T_0) \tag{13}$$

## 4. Results and Discussion

The B-scan image of the defects by LGU inspection is shown in Figure 5a. The indications marked in the red frames are the defects of 0.1, 0.2, and 0.3 mm in depth. However, the B-scan image of LGU cannot provide accurate depth information due to the wide time-domain signal modulated by the defect and sample surface. The C-scan image of the LGU detection of the pre-made six defects is shown in Figure 5b. All defects are detected with depths of 0.1, 0.2, 0.3, 0.5, 0.7, and 1 mm. The horizontal position of the defects could be located by measuring the edge of these indications. But it's still difficult to accurately measure the depth of defects, as there is only qualitative evidence that the signal strength becomes weaker as the defect depth increases. However, the C-scan image is helpful to find and extract the defect's A-scan signal for further analysis.

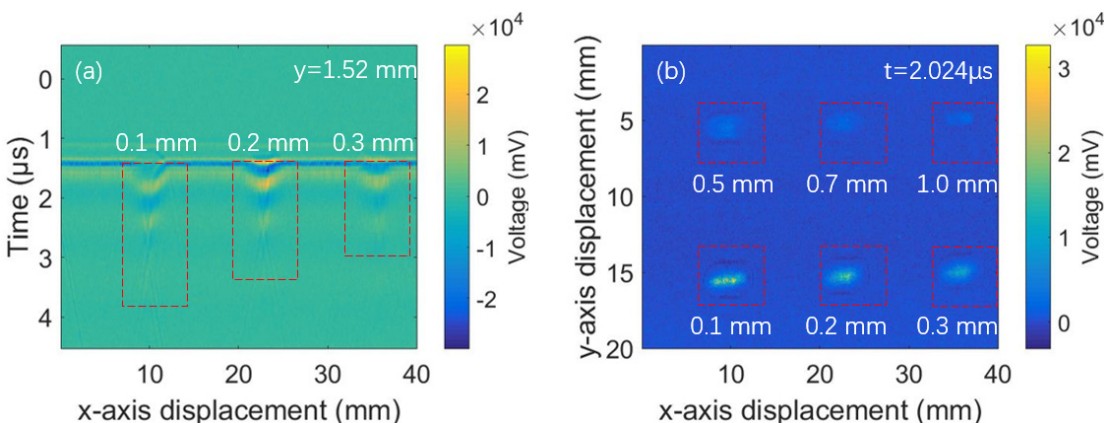

**Figure 5.** (**a**) B-scan and (**b**) C-scan plots of LGU inspection of defects.

Figure 6a–f are the A-scan signals of defects with six different depths extracted from the horizontal position provided by the C-scan image in Figure 5b. The black line in the figure represents the time domain A-scan signal with amplitude and time, and the blue line represents the spectrum domain signal with frequency and voltage obtained by the fast Fourier transform (FFT). Comparison of the A-scan signals reveals that the width of the signals decreases as the defect depth increases. The embedded notch of 0.1 mm and the sample surface forms a foil-like structure and induces typical Lamb waves signals, as shown in Figure 6a. When the depth of the notch becomes larger, the LGU signals are only modulated by the sample surface and induce typical surface waves signals, as shown in Figure 6f. It means that the mode conversion happens when the defects exist and the depth changes. Comparison of the spectrum domain signals shows a frequency shift, which occurs when the defect depth changes. Double frequency peaks appear when the Lamb wave and the surface wave coexist, as shown in Figure 6d, e. Further time-frequency

analysis by means of the Wavelet-transform is introduced to distinguish the Lamb wave mode and quantify the correlated central frequency and time.

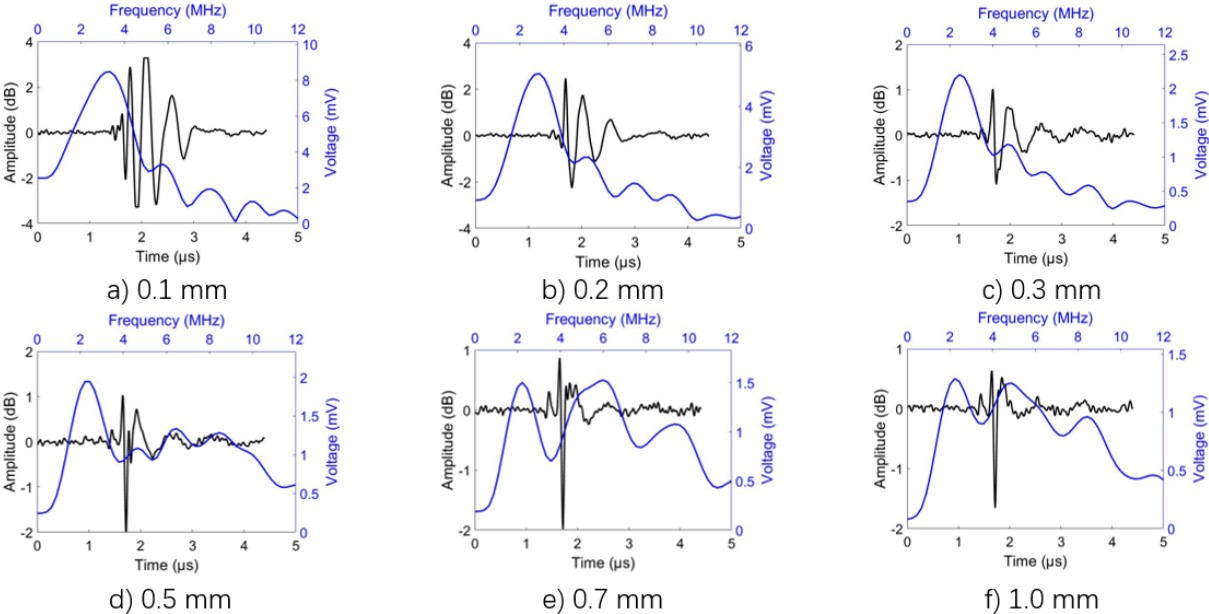

**Figure 6.** (**a**) 0.1 mm, (**b**) 0.2 mm, (**c**) 0.3 mm, (**d**) 0.5 mm, (**e**) 0.7 mm, and (**f**) 1.0 mm A-scan signal of six different depth defects and the spectrum signals obtained after Fourier transform.

Figure 7a–f are the time-frequency images of the six A-scan signals obtained by the wavelet-transformation. The yellow area represents the location of a wavelet energy concentration. It can be clearly seen from the figures that, as the depth increases, the wavelet energy packet gradually shifts from low to high frequency. This is because, when the depth is small, the Lamb wave is the main signal form. When the depth is 1 mm, the surface wave is the main signal form. However, the low-frequency wavelet energy concentration of the Lamb wave can still be seen. The time-frequency images also prove the frequency shift and mode conversion of LGU waves. The propagation time of the Lamb wave can be accurately recorded if the wave mode and frequency are fixed. Then, the velocity of the Lamb wave can be calculated using the Formulas (6)–(10).

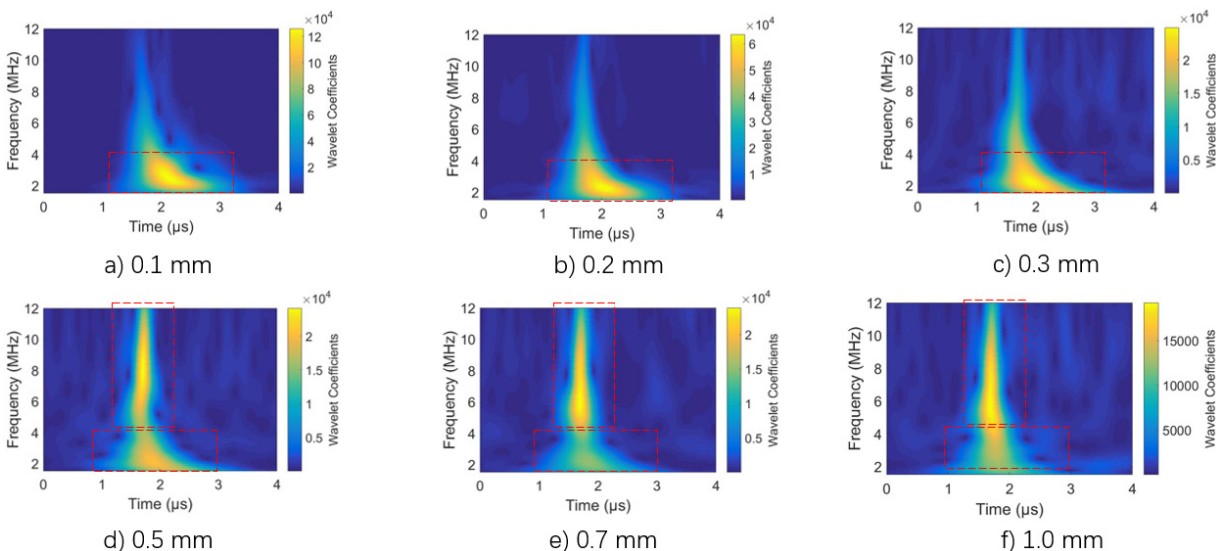

**Figure 7.** Time—frequency images of six defects with different depths of of (**a**) 0.1 mm, (**b**) 0.2 mm, (**c**) 0.3 mm, (**d**) 0.5 mm, (**e**) 0.7 mm, and (**f**) 1.0 mm obtained by the A-scan wavelet-transform.

The theoretical and experimental results of the relationship between propagation velocity and defect depth are shown in Figure 8. In order to find the best matching curve for calibration, five frequencies of the low-frequency wavelet energy packets have been extracted for measurement accuracy evaluation. In Figure 8, the trends of defect depth and sound velocity curves measured from the experimental results are close to the theoretical value. It can be seen that, as the depth of the defect increases, the sound velocity increases, and the phase velocity is very close to the Rayleigh wave velocity when the depth exceeds 1 mm. The difference between the theoretical and experimental results is the smallest when the frequency is 2.2 MHz and the fitting coefficient reaches 0.98. This is because 2.2 MHz is the center frequency of the wavelet energy packet.

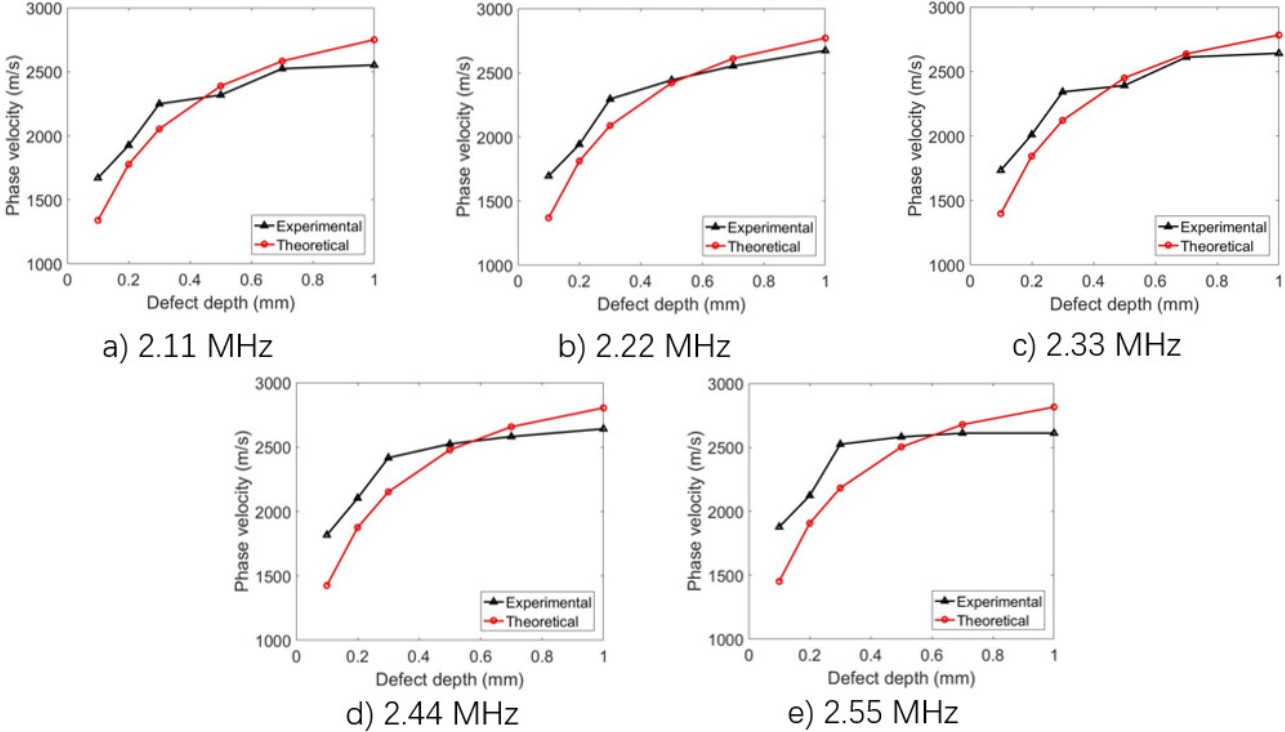

**Figure 8.** Theoretical and experimental relationship between phase velocity and defect depths at different frequencies: (**a**) 2.11 MHz, (**b**) 2.22 MHz, (**c**) 2.33 MHz, (**d**) 2.44 MHz, (**e**) 2.55 MHz.

Figure 9 shows the depth measurement results of the LGU detection of AM defects mentioned above. The curve correlation coefficient between the designed and measured values is 0.983. The result indicates that it is feasible to measure defect depth based on the dispersion characteristics and wavelet-transform of LGU signals. According to the extracted frequency and sound velocity, the defect can be accurately measured in depth. However, when the defect depth is too large, the main form of the ultrasonic wave is the Rayleigh wave. The energy ratio of the Lamb wave is small, as shown in the time-frequency image (Figure 7f), resulting in an error of 20% when the depth reaches 1 mm. The recommended defect depth range for accurate measurement is suggested to be lower than 0.8 mm, which is enough to meet the inspection layers thickness of AM methods, such as the selective melting method. The accurate position provided by the proposed method in this paper would be helpful for repairing the defective part rapidly and improving the printing efficiency and printing performance of additive/subtractive manufacturing methods.

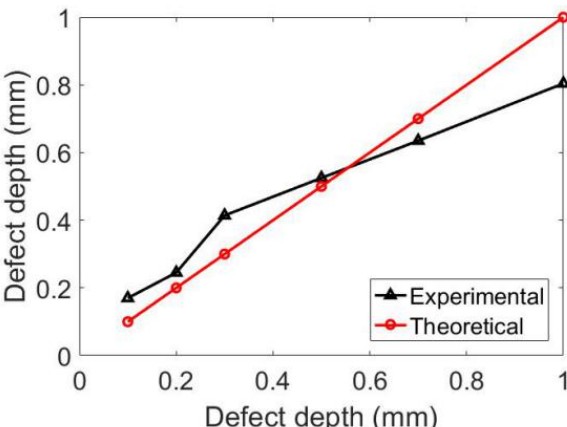

**Figure 9.** Comparison chart of the experimental and theoretical value of the effect depth.

## 5. Conclusions

In this paper, a mode-conversion phenomenon from LGU surface waves to Lamb waves caused by subsurface defects at different depths is observed and systematically explored using LGU testing experiments. A novel method to measure the depth of subsurface defects is proposed based on the Lamb waves velocity dispersion analysis by the wavelet-transform. The conclusions are as follows:

(1) The mode-conversion is attributed to the velocity dispersion of the LGU. The central frequency and propagation velocity of the laser-induced surface wave are changed as the depth of the defects change.

(2) The measured result of defect depth is very close to the theoretical value with a fitting coefficient of 0.98. The recommended defect depth range for accurate measurement is suggested to be lower than 0.8 mm, which is enough to meet the inspection layers thickness of AM methods, such as the selective melting method.

In further work, we will consider adding material samples or exciting single-frequency Rayleigh waves for more accurate measuring of the depth of subsurface defects.

**Author Contributions:** Conceptualization, Z.X.; methodology, W.X.; software, J.Z. and W.X.; validation, Z.X.; formal analysis, Z.X.; investigation, Z.X.; resources, J.Z. and Y.P. and M.W. and B.Y.; data curation, Z.X.; writing—original draft preparation, Z.X.; writing—review and editing, J.Z. and V.P. and B.Y.; visualization, Z.X.; supervision, J.Z. and W.X. and V.P.; project administration, Z.X.; funding acquisition, J.Z. All authors have read and agreed to the published version of the manuscript.

**Funding:** This work was supported by the National Key R&D Program of China (Grant No. 2018YFB1106100) and the Fast Support Project for Installation and Development (Grant No. 80904010502).

**Institutional Review Board Statement:** Not applicable.

**Informed Consent Statement:** Not applicable.

**Data Availability Statement:** Data Sharing is not applicable for this article.

**Conflicts of Interest:** The authors declare no conflict of interest.

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
