# Peer review of "Measuring the Depth of Subsurface Defects in Additive Manufacturing Components by Laser-Generated Ultrasound"

_metals, doi:10.3390/met12030437_

Round 1

Reviewer 1 Report

The novelties brought in the paper and/or the originality of the research in this study are not visible or have not been presented as such. I would like authors to extend the conclusions with the requested arguments.
I recommend refining the language used and  rewriting the manuscris  more clearly in technical English, by forming more concrete sentences from a scientific point of view.

Reviewer 2 Report

This manuscript describes the measurement of the depth of subsurface defects in additive manufacturing components by laser-generated ultrasound. This is a very interesting work in which the authors study the mode-conversion from laser-generated surface waves to Lamb waves caused by subsurface defects at different depths, to then manufacture a 316L stainless steel sample in order to validate the method. The paper is well structured, featuring a good introduction. The theoretical model regarding the velocity dispersion of the Lamb waves and the velocity calculation is correct, to the best of my knowledge. The figures summarize the key aspects of the work in a proper graphical way. The results are extremely relevant for robust and reliable defect depth measurement in industrial applications, since the fitting coefficient between the measured result and the designed one is 0.98 and the recommended defect depth range for high accuracy is lower than 0.8mm. Finally, the Conclusions are perfectly based on the previous findings. All in all, this is an impressive work and a perfect match for MDPI Metals. Therefore, I recommend its acceptance for publication in its present form.

Author Response

Thank you for your summary. We appreciate the reviewer’s positive evaluation of our work.

Reviewer 3 Report

The Manuscript deals with defect monitoring and detection through a Lamb waves velocity dispersion analysis. The research topic has a high scientific and technological relevance. 

The scope of the Manuscript and the followed methodology are well introduced and discussed. Nevertheless, some flaws must be solved before paper publishing: 

  • several typos and format errors have been observed in the Manuscript. Please, solve them, improving the quality of the text;
  • all the symbols used in equations must be introduced and discussed in the text (not only in the Tables). Please, improve them.
